# Traj2Former: A Local Context-aware Snapshot and Sequential Dual Fusion Transformer for Trajectory Classification

Yuan Xie
National University of Singapore
yuan_xie@nus.edu.sg

Yichen Zhang
National University of Singapore
yichenz@nus.edu.sg

Yifang Yin
Institute for Infocomm Research (I²R), A*STAR
yiny@i2r.a-star.edu.sg

Sheng Zhang
Institute for Infocomm Research (I²R), A*STAR
zhangs2@i2r.a-star.edu.sg

Ying Zhang
Northwestern Polytechnical University
izhangying@nwpu.edu.cn

Rajiv Ratn Shah
IIIT-Delhi
rajivratn@iiitd.ac.in

Roger Zimmermann
National University of Singapore
rogerz@nus.edu.sg

Guoqing Xiao*
Hunan University
xiaoguoqing@hnu.edu.cn

## Abstract

The wide use of mobile devices has led to a proliferated creation of extensive trajectory data, rendering trajectory classification increasingly vital and challenging for downstream applications. Existing deep learning methods offer powerful feature extraction capabilities to detect nuanced variances in trajectory classification tasks. However, their effectiveness remains compromised by the following two unsolved challenges. First, identifying the distribution of nearby trajectories based on noisy and sparse GPS coordinates poses a significant challenge, providing critical contextual features to the classification. Second, though efforts have been made to incorporate a shape feature by rendering trajectories into images, they fail to model the local correspondence between GPS points and image pixels. To address these issues, we propose a novel model termed Traj2Former to spotlight the spatial distribution of the adjacent trajectory points (*i.e.*, contextual snapshot) and enhance the snapshot fusion between the trajectory data and the corresponding spatial contexts. We propose a new GPS rendering method to generate contextual snapshots, but it can be applied from a trajectory database to a digital map. Moreover, to capture diverse temporal patterns, we conduct a multi-scale sequential fusion by compressing the trajectory data with differing rates. Extensive experiments have been conducted to verify the superiority of the Traj2Former model.

## CCS Concepts

• **Information systems** → **Geographic information systems**; • **Computing methodologies** → **Supervised learning by classification**.

## Keywords

Trajectory Classification, Transformer, Mapped Trajectory.

*Corresponding Author

**ACM Reference Format:**
Yuan Xie, Yichen Zhang, Yifang Yin, Sheng Zhang, Ying Zhang, Rajiv Ratn Shah, Roger Zimmermann, and Guoqing Xiao. 2024. Traj2Former: A Local Context-aware Snapshot and Sequential Dual Fusion Transformer for Trajectory Classification. In *Proceedings of the 32nd ACM International Conference on Multimedia (MM '24), October 28-November 1, 2024, Melbourne, VIC, Australia.* ACM, New York, NY, USA, 9 pages. https://doi.org/10.1145/3664647.3681340

## 1 Introduction

The widespread use of mobile devices has led to the production of vast amounts of trajectory data, typically in the form of two-dimensional GPS point sequences. Classification is a crucial and fundamental challenge within the domain of spatio-temporal analytics. Applications of trajectory classification extend to various downstream tasks, such as providing trip recommendations [22] and enhancing smart transportation systems [2, 24, 37].

Existing research on trajectory classification can be roughly divided into trajectory-based methods, image-based methods, and their fusion. For trajectory-based methods, early work manually extracts speed, acceleration, and bearing as features [6] and employs traditional machine learning models such as SVM, Bayes, KNN, and Random Forest as the classifier [36]. Recent efforts mostly adopt deep learning models such as CNN and RNN [6, 27] to learn more robust trajectory representations from large-scale training datasets. For example, Liang *et al.* proposed TrajODE [20] to capture the representation of the continuous-time dynamic in trajectory data inherently. Further, they proposed TrajFormer [19] to encode the spatio-temporal intervals of the continuous trajectory sequences to deal with GPS noise and sparsity issues. Meanwhile, a parallel research direction for trajectory classification is to render trajectories into images [8, 13]. This strategy captures the trajectory shape information, constrained by the underlying road network and thus complementary to the traditional trajectory representation. For instance, the Estimator model [13] was proposed to transmute the target trajectory into an image, and then fuse it with the original trajectory data to achieve performance gains. Yuki *et al.* [8] extracted features from the formulated image by the trajectory for the transportation mode estimation. Moreover, Kontopoulos *et al.* [14] proposed TraClets to fine-tune the VGG16 model on the trajectory image features for the trajectory classification. However, these methods primarily focus on rendering a single target trajectory without fully exploring other trajectories in a large historical

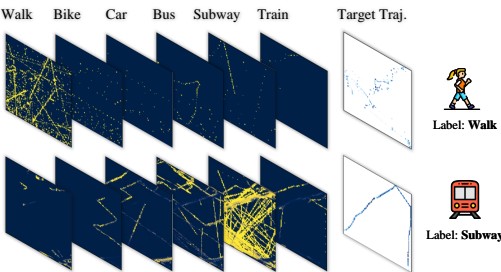

**Figure 1: The distribution of trajectories with different transportation modes in the neighborhood of a target trajectory.**

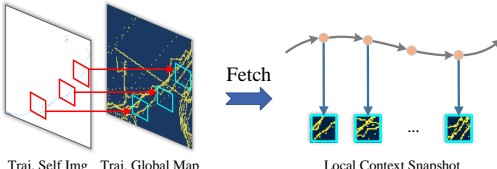

**Figure 2: Illustration of fetching local contexts from the global map and their correspondence with the GPS points.**

database. Furthermore, the fusion of trajectory- and image-based features is mostly performed before the output layer without exchanging information locally.

As illustrated in Fig. 1, the historical trajectory distribution is an important feature in revealing the transportation mode of the target trajectory. For example, if the nearby samples of a target trajectory are majorly labeled as "Walk" rather than "Car" or "Train", then the target trajectory is more likely to follow a sidewalk that is highly correlated with the label "Walk" rather than a highway which is high related to "Car". However, such semantic information [25] has not been fully investigated in previous work.

Motivated by the above observations, we propose a novel framework termed Traj2Former, which first transforms trajectories into global maps and then performs local context-aware snapshots and multi-scale sequential dual fusion. Specifically, our framework extracts a comprehensive global map enhanced by semantics through the rendering of a large-scale historical trajectory database. This conversion leads to the creation of two distinct maps: 1) a physical feature map, which captures the concrete environmental characteristics, and 2) a class distribution map, which depicts the probabilistic spread of different categorical elements across the global map. Thereafter, we propose to extract local contextual snapshots (see Fig. 2) from the global map to reduce irrelevant information, which is next enhanced by a novel transformer-based map encoder to further reduce the data noise. The contextual representation generated by the map encoder and the corresponding trajectory representation generated by the trajectory encoder are next fused hierarchically by progressively condensing the input trajectory, aiming to capture both neighborhood and sequential patterns in multiscale. Finally, we would like to emphasize that our proposed Traj2Former is a unified fusion framework, which is capable of accommodating maps generated from different data sources, *e.g.*, trajectory database, crowdsourced map, or satellite imagery. Extensive experiments have been performed on two public benchmark

datasets, namely Geolife and MTL, to evaluate our model design and effectiveness. Here we summarize our contributions as follows.

- To our knowledge, we are the first to present a unified fusion framework that performs local context-aware snapshot fusion by simultaneously modeling multiscale correlations between trajectory and neighborhood representations.
- We propose to generate an enhanced global map by rendering from a large-scale trajectory database to support contextual snapshot extraction, which provides vital and complementary information to the traditional trajectory representations.
- We introduce a new and general Transformer-based Map Encoder, which is capable of coping with diverse map sources, to further reduce the impact of noise in crowdsourced data.
- We compare our model to the state-of-the-art trajectory classification methods, where significant performance gain has been obtained on both Geolife and MTL benchmark datasets.

## 2 Related Work

We review related works which can be roughly categorized into point-aware and image-aware trajectory classification methods.

**Point-aware Trajectory Classification.** A trajectory can be represented via multiple data formats [28–30, 34]. The usual formats of trajectories are comprised of a sequence of 2-dimensional GPS points, *e.g.*, spatial locations, and temporal timestamps. Point-aware methods [1, 7, 17] are proposed to capture the spatio-temporal features [16] among the trajectory that is comprised of the separated GPS points. Zheng et al. [36] proposed four different methods including KNN [11], Bayes [21], SVM [4] and Random Forest [10], to classify a user's transportation mode. Furthermore, Zheng et al. [35] proposed a supervised learning approach to identify the sophisticated features and designed a graph-based postprocessing algorithm to further improve the inference performance. Additionally, Liang et al. [19] adapted transformers to model trajectories to embed the spatio-temporal intervals of the continuous trajectory points and squeeze the points to speed up the representation learning. Lee et al. [18] explored the region- and the trajectory-based features to overcome the discriminative parts of the trajectory identification. However, these methods cannot easily capture the coarse-grained spatial features and the nearing distribution of the trajectories.

**Image-aware Trajectory Classification.** To boost classification performance, some existing studies focus on mapping trajectories to images to capture the shape of the objective trajectory. Hu et al. [13] developed CNN-TCN to utilize the shape of a trajectory and the time embedding for the trajectory classification. Yuki et al. [8] converted a trajectory to a grayscale image and integrated it with hand-extracted features to classify trajectories. Kontopoulos et al. [15] identified a vessel's mobility patterns by fusing the vessel's trajectory and the image of the corresponding trajectory. Furthermore, Kontopoulos et al. [14] proposed TraClets that fine-tune the VGG16 for image-based trajectory classification. These existing approaches directly introduce deep learning methods to fuse the trajectory embedding [9] with the mapped trajectory images to solve the trajectory classification problem. However, the semantic distribution information of nearby trajectories has not been fully investigated in previous works.

## 3 Problem Formulation

**Problem Statement:** With a set of trajectories $T=\{T_1, T_2, \cdots, T_n\}$ compiled from moving objects, the task of trajectory classification involves categorizing these sequences into distinct groups, such as walk, car, and other modes of transport. A unified fusion network (Section 4) is proposed to effectively aggregate the information from both the trajectory input and the map input. Different from previous work, we propose to leverage a historical trajectory database to generate a comprehensive global map enriched by semantics (Section 5). The trajectory input is extracted from the original GPS format. The map input is a multi-channel image containing contextual information, which can be generated from either historical trajectories or crowdsourced online maps (*e.g.*, OpenStreetMap).

DEFINITION 1. *(Trajectory.) A trajectory T contains a time-order sequence GPS points, i.e., $T=\langle p_1, p_2, \cdots, p_n \rangle$. Each GPS point $p_i (1 \le i \le n)$ is represented by $\{\langle lat_i, lon_i \rangle, t_i\}$, where $lat_i$ is the latitude location, $lon_i$ is the longitude location, and $t_i$ is the timestamp.*

DEFINITION 2. *(Historical Trajectory Database.) The database contains all historical trajectories in the training set with trajectory labels. The item in the database is presented as $T_j=\{\langle p_1, p_2, \cdots, p_n \rangle, mode_j\}$, where $mode_j$ signifies the transport label.*

DEFINITION 3. *(Self Image.) The self-image of the target trajectory is a multi-channel image. Each channel consists of $W \times H$ grid cells and represents a feature map such as velocity and acceleration.*

DEFINITION 4. *(Global Map.) The global map with $W \times H$ resolution is generated by rendering adjacent trajectories from the historical trajectory database to a multi-channel image. It yields substantial advantages by providing background information on the transportation in the neighborhood.*

## 4 Traj2Former Architecture

The overview framework of our proposed Traj2Former model is illustrated in Fig. 3, which consists of 1) a Trajectory Encoder that encodes the input trajectory with robust point-level representations; 2) a Local Context-aware Fusion Network that effectively fuses trajectory and map representations with varying spatial granularity; and 3) a Transformer-based Map Encoder that enhances the local contexts and updates the global map with self-attentions.

### 4.1 Trajectory Encoder

By using the Trajectory Encoder as an initial step, the Traj2Former model gains a refined initial representation of the trajectory data. This improves its performance during the subsequent integration with image-based trajectory data.

*4.1.1 Sequential Trajectory Feature Extraction.* Following previous work [6], we extract velocity, acceleration, jerk, heading, and its changing rate from GPS trajectories, which are the key features for training an effective trajectory classifier. To avoid obtaining unstable features for each track point, we implement a moving average window to approximate the features at each point by moving through the whole trajectory. The time span of the sliding window is set to $\triangle t$. For a sample point $i$, we set it to be the starting point of the window and calculate the distance $d$ from the first sample point to the last sample point. Then, the velocity, acceleration, and

jerk of sample point $i$ are computed as $v_i = d/\triangle t$, $r_i = |v_j - v_i|/\triangle t$, and $a_i = |r_j - r_i|/\triangle t$, respectively. Similarly, we also compute the heading and its changing rate at each track point.

*4.1.2 Sequential Trajectory Encoding.* The extracted features described above are then encoded into a high-dimensional vector space using a sophisticated CNN-based neural network [6]. This learned trajectory embeddings comprise point-level robust representations, which will be fed into the Trajectory Compressor to generate informative segment-level representations. Formally, the embedding of a trajectory $T_j$ is expressed as:

$$E_p = f_{emb}(T_j) \in \mathbb{R}^{\xi}, \tag{1}$$

where $f_{emb}$ refers to the trajectory encoder, and $\xi$ denotes the dimensionality of the feature space. By embracing the Trajectory Encoder as a preliminary step, the Traj2Former model benefits from a nuanced initial representation of the trajectory data, leading to improved performance and accuracy in the subsequent fusion with spatial trajectory features.

### 4.2 Local Context-aware Multi-scale Fusion

Though efforts have been made on fusing the sequential and spatial trajectory features [13], the modeling of the local correspondences is largely overlooked. Therefore, we propose a novel fusion method to highlight both global and local fusion at a multi-scale level.

*4.2.1 Global Segment-to-Image Fusion.* A naive fusion approach involves directly merging the self-image feature representation with the trajectory embedding [13] to potentially improve classification outcomes. The mathematical formulation of this coarse-grained naive fusion technique can be described as follows:

$$F_{s2i} = E_p \otimes E_m, \tag{2}$$

where $E_p$ represents the trajectory embedding defined in Eq. 1, $E_m$ signifies the spatial trajectory embedding which will be introduced in Section 5.1.2, and $\otimes$ is the concatenation operation. By merging different forms of data representations, the global segment-to-image fusion approach generates a richer and more comprehensive feature set. Moreover, merging at the coarse-grained level allows for better flexibility in the development process, such as optimizing or modifying each component independently. However, naive fusion overlooks the local correlations between different data modalities and limits the model's ability to capture the interactions on fine-grained level features, which could lead to a decrease in performance.

*4.2.2 Local Point-to-Pixel Fusion.* To explore the fine-grained information, we propose a fusion method that primarily aggregates the point-level representations with the corresponding local image representation, shown as follows,

$$F_{p2p} = L\_Fusion(\langle e_1^p \otimes e_1^m \rangle, \cdots, \langle e_n^p \otimes e_n^m \rangle), \tag{3}$$

where $e_1^p, \cdots, e_n^p$ are the point-wise embedding from the raw trajectory, $e_1^m, \cdots, e_n^m$ are the local spatial trajectory representation, and $n$ is the length of the target trajectory. $L\_Fusion$ is a general fusion method with mean-pooling, max-pooling, or other fusion methods to aggregate the features. Here, we adopt concatenation as the fusion method to generate the segment-level embedding. Point-to-pixel fusion exploits the correlation among fine-grained

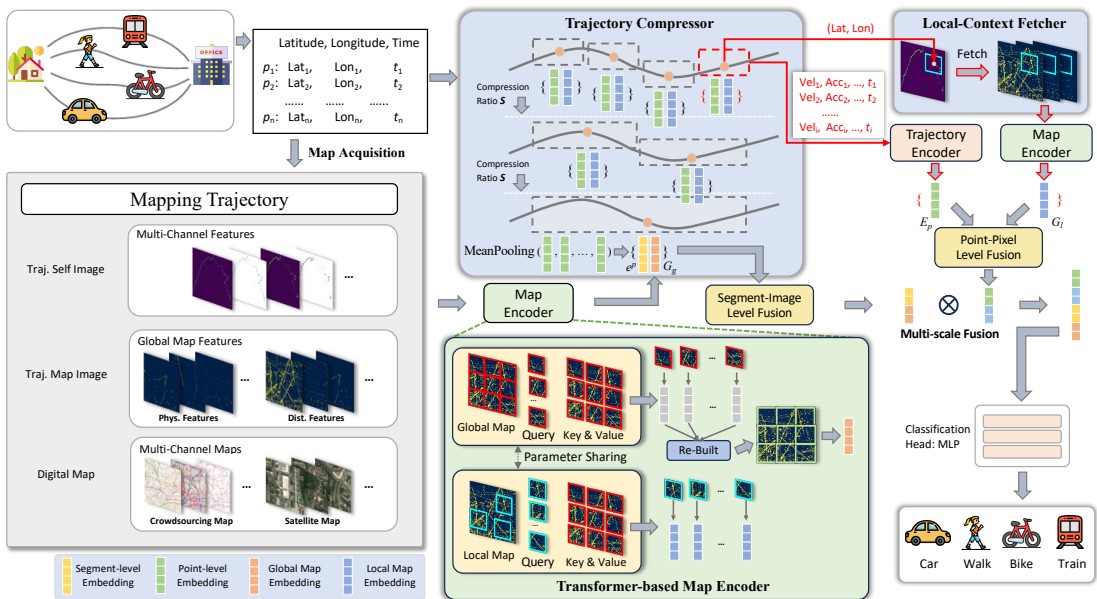

**Figure 3: System overview of our proposed Traj2Former framework.**

features that are overlooked in segment-to-image fusion. It also extracts consistent features across multiple data sources to gain a deeper understanding of the contextual information. However, multiple points in a trajectory may correspond to the same pixel in the map, resulting in duplicate local contexts retrieved in the point-wise fusion. These redundant local contexts offer minimal new or valuable information and should therefore be excluded from further processing to improve the system's efficiency.

*4.2.3 Efficient Hierarchical Global-Local Fusion.* To simultaneously harness the benefits of both global and local fusion, we propose a new Hierarchical Global-Local (HGL) Fusion approach, as depicted in Fig. 3. To avoid extracting redundant local information, we innovatively develop a Trajectory Compressor (TC) module and a Local-Context Fetcher (LCF) module to support efficient point-pixel level fusion. Thereafter, the global map and trajectory embedding are fused in the segment-image level fusion module. Coarse-grained features can enhance the final representation by offering information from a global perspective. Finally, the outputs of point-to-pixel fusion at different scales and the outputs of segment-to-image fusion are concatenated to obtain a final representation $F_{dual}$, which is formally given as

$$F_{dual} = \hat{F}_{p2p} \otimes F_{s2i}, \ \hat{F}_{p2p} = F_{p2p}^1 \otimes \cdots \otimes F_{p2p}^L \tag{4}$$

where $\hat{F}_{p2p}$ denotes the multi-scale point-to-pixel fusion across $L$ layers and $F_{s2i}$ is the segment-to-image fusion as shown in Eq. 2. **Trajectory Compressor.** To prevent generating redundant information and incurring higher time costs when extracting local maps from the global map, the initial trajectory is compressed to obtain a segment-level feature concentrated with essential information. Particularly, we divide the trajectory into $S$ (referred to as the compression ratio) segments recursively, and extract the segment-level representations based on the average pooling of the features from the previous level. Thereafter, we project the location of the center

point in each segment to the image coordinates on the map. These pixel coordinates will be passed to the LCF to extract local context from the map. We also propose a novel Transformer-based Map Encoder (see Section 4.3) to extract complementary spatial trajectory features from the local context, which are next fused with the segment-level trajectory representations.

**Local Context Fetcher.** As discussed in the introduction, the goal of this module is to reduce the noise and irrelevant information presented in the global map. To achieve a high concentration of local information, we adopt a Local Context Fetcher (LCF) module guided by the compressed trajectory. Precisely, for each pixel that correlates a trajectory segment, we crop a sub-image on the global map with a size of $c_p \times c_p$ centered at respective pixel coordinates, which can be expressed as

$$G_l = LCF(G_g, \{i, j\}, c_p), \tag{5}$$

where $G_l$ and $G_g$ represent the local and global contexts, $LCF$ denotes the Local Context Fetcher module, $c_p$ designates the crop size, and $\{i, j\}$ signifies the pixel coordinates. The local contexts are tailored to offer the distinct advantage of mitigating noise while maintaining a sharp focus on the relevant spatial context.

## 4.3 Transformer-based Map Encoder

To cope with maps generated from different sources, we would like to design a generalized map encoder with the capability of extracting robust features regardless of the map type. Different types of maps may have their own limitations. For example, the global map generated from historical GPS trajectories may suffer from GPS intrinsic noise and sparsity. OpenStreetMap is crowdsourced and thus may contain duplicate or inaccurate information. Motivated by the above observations, we propose a Transformer-based Map Encoder to aggregate information from other regions to enhance the point-level fusion and segment-level fusion.

*4.3.1   Local Context Refinement for Point-to-Pixel Fusion.* To refine the local context for enhancing the point-level fusion, we adopt the multi-head attention model [26] to train the local context map embedding. Especially, we process the two-dimensional global map image and reshape it from $\mathbb{R}^{C \times H \times W}$ into a sequence of two-dimension patches $p_i \in P, P \in \mathbb{R}^{N \times (C \cdot r^2)}$, where $H \times W$ is the resolution of the global map, $r^2$ is the resolution of every single patch, $C$ is the number of feature channels, and $N$ is the number of patches (e.g., $N = HW/r^2$). Next, the patches are flattened to $d$-dimensional vectors with a trainable linear projection $p_i \in \mathbb{R}^{N \times d}$. The traditional attention mechanism is represented as

$$MultiHead(Q, K, V) = Concat(head_1, \cdots, head_h)W^o,$$
$$head_i = Attention(QW_i^Q, KW_i^K, VW_i^V),$$

where the local maps are the query, and the flattened global maps are the key and value in our applied multi-head attention module. More specifically, the query is given as $Q \in \mathbb{R}^{v \times (C \cdot r^2)}$ and the key and value are defined as $K = V \in \mathbb{R}^{N \times (C \cdot r^2)}$, where $v$ is the number of local contexts returned by the Local Context Fetcher.

*4.3.2   Global Map Enhancement for Segment-to-Image Fusion.* To refine the global maps and extract highly representative features, we employ a multi-head self-attention model to iteratively update the global map at each layer within the Traj2Former framework. The global map enhancement is proposed to enhance the segment-level fusion. For the self-attention model, the query, the key, and the value share the same sequence of patches from the global map. The relationship is defined as $Q = K = V \in \mathbb{R}^{N \times (C \cdot r^2)}$. For optimization efficiency and feature consistency, the multi-head self-attention model shares the same parameters of the local context refinement module introduced in Section 4.3.1. The same transformative weights are applied when updating the global map and extracting the local contexts. Thus, the global map and local contexts are mutually enhanced, benefiting from the consistent set of learned features.

## 4.4   Objectives and Optimization

Our Traj2Former model consists of $L$ layers where MLPs are applied as decoders to convert trajectory representations to class distributions. The $L^{th}$ layer produces the final predictions. Building on the observation that sub-trajectories share the same class label, we train our Traj2Former model by jointly minimizing the cross-entropy loss on all layers as given below

$$\mathcal{L} = \alpha \cdot \sum_{i=1}^{L-1} \mathcal{L}_i(\hat{y}, y) + \beta \cdot \mathcal{L}_L(\hat{y}, y), \tag{6}$$

where $\hat{y}$ and $y$ are the predicted class and true label, respectively. $\mathcal{L}_i$ is the cross-entropy on output of the $i$-th layer and $\mathcal{L}_L$ is the cross-entropy on the final output of the last layer.

## 5   map acquisition

Features such as speed, acceleration rate, and bearing play a decisive role in trajectory classification [6]. To obtain complementary information from the historical trajectories, we extract not only the aforementioned physical statistics but also the transportation mode class distributions to generate a global map in our work.

## 5.1   Spatial Trajectory Feature Extraction

Efforts have been made to convert GPS trajectories [13, 32] and crowdsourced map data [31] into images, enriching trajectory data with visual information (see Section 4.1.1). While sequential features analyze movement patterns over time, trajectory images highlight the distribution of nearby trajectories, offering insights into the road environment. The model name, Traj2Former, reflects this dual approach, encompassing both sequential and spatial features.

*5.1.1   Self-image generation from a single trajectory.* To enable visual representation of the trajectory in a more intuitive and accessible format, the initial trajectory is converted to a self-image by transforming GPS data points into pixel coordinates in an image. We compute the latitude and longitude span of the whole trajectory and obtain the pixel index of each GPS point by setting the image size to $W \times H$. In particular, for each sample point $\{\langle lat_i, lon_i \rangle, t_i\}$, we compute the pixel index as follows,

$$lat_i^p = \frac{lat_i - min_a}{W}, \ lon_i^p = \frac{lon_i - min_o}{H}, \tag{7}$$

where $min_a$ and $min_o$ are the minimal latitude and longitude of the trajectory, $lat_i^p$ and $lon_i^p$ are the obtained pixel index. Based on the given trajectory, we extract five important features, including speed, acceleration, angle, acceleration rate, and angle rate. Since multiple GPS points can be projected to the sample pixel on the map, we compute their average as the final feature.

*5.1.2   Global map generation from a historical database.* To create a global map with nearby trajectories, we aggregate trajectories from the historical database and apply the self-image formulation method, replacing the individual trajectory with a set of neighboring trajectories. Two types of maps are converted from the historical trajectory database: physical statistic maps and class distribution maps, collectively called global maps. The region covered by the global map is defined to be the same as the self-image introduced above. Subsequently, we retrieve all trajectories within that region from the historical database to generate the global map. Besides, we have enriched the original features by incorporating the detailed statistical descriptors of speed, acceleration, and angle, specifically the maximal, minimal, and average values of each attribute, and the number of GPS points. The physical statistic features $C_p$ are formulated based on channel-wise concatenation as,

$$C_p = C_f^1 \odot \cdots \odot C_f^m, \tag{8}$$

where $C_f^i$, $1 \leq i \leq m$, is the extracted features from the adjacent trajectories, and $m$ is the number of physical feature channels. However, the physical statistic features neglect the class distribution of the neighboring trajectories, which intuitively captures the road environment. To enrich the global map features, we further generate the class distribution features as one component of the global map. Assume that there are $k$ classes in the dataset, then $k$ channels will be created, each of which records the normalized count of the GPS points belonging to a specific class. Formally, we have

$$C_d = C_s^1 \odot \cdots \odot C_s^k, \tag{9}$$

where $C_s^i$ is the distribution channel for the $i$-th class, $1 \leq i \leq k$. The final global map representation $E_m$ is comprised of the physical feature map and the class distribution map as $E_m = C_p \odot C_d$.

## 5.2 Public Map Sources

Public map sources can also be utilized in our Traj2Former model, where APIs are usually provided for downloading map visualizations such as digital map [33] and satellite map [32].

*5.2.1 Digital Map.* In real-world scenarios, we would like to leverage additional map resources that are publicly available. This is important since our model has the ability to fuse different map features from multiple map sources to boost classification performance. Digital maps, such as HERE map and OpenStreetMap, can be rendered in different styles (*i.e.*, Cycling Map, Transportation Map, Topography Map, and Humanitarian Relief Map), which capture different road attributes to further enhance the model performance.

*5.2.2 Satellite Imagery.* Our Traj2Former model exhibits remarkable flexibility, proving effective not only with self-generated global maps but also with satellite imagery. To assess the performance and reliability of Traj2Former when applied to satellite maps, the HERE map API is utilized. This tool enables us to precisely extract satellite images corresponding to target trajectories. Table 6 presents the results of the ablation study on using different map sources.

## 6 Experiments

### 6.1 Experimental Settings

Existing works divide trajectory data into segments by counting a specific number of valid GPS points [5]. However, the time duration of each segment is critical in practical applications, as the transportation mode typically remains constant over a continuous time span. To achieve this, we inserted a new point at one-second intervals between two consecutive points, resulting in segments composed of 600 points (*i.e.*, 10 minutes) in Geolife and inserted a new point at five-second intervals of 650 points in MTL.

**Geolife and MTL datasets.** The Geolife dataset contains 27,795 trajectory segments and six transportation modes. The MTL includes four selected modes of transportation and contains 23,406 segments. These two datasets, with a division of training and testing data, were maintained at an 8:2 ratio across each transportation mode category. Details of the dataset are illustrated in the Appendix.

**Implementation Details.** We apply the Adam optimizer during training and adopt a learning rate of 0.001 with a weight decay of 0.001. The batch size is set to 64 and all generated images are 50×50 in size. The kernel size for the sequential and spatial trajectory embedding is set to 3 and 6, respectively. The Traj2Former model has 3 layers, and we set $\alpha$=0.3, $\beta$=0.4 in the loss function. All experiments are conducted on a single NVIDIA RTX A6000 with 48GB memory.

**Metrics.** For the trajectory classification problem, we follow [3] to report the sub-class classification accuracy and the overall accuracy to evaluate our model's effectiveness. Here, the accuracy is the ratio of the number of correct samples to the number of the whole samples. Besides, we also list the per-class accuracy to show the detailed effectiveness of each class. In the presented tables, we also use the notation $\Delta$ to indicate the relative improvement of accuracy compared with the state-of-the-art method.

## 6.2 Comparison to the State-of-the-art Methods

We compared our proposed method to the following state-of-art methods and report the results in Table 1.

- **SVM, KNN, RF** [36] applies the machine learning methods to solve the trajectory prediction problem.
- **RNN, GRU, STGRU** [3, 27] focuses on solving the variable length sequence to address the constraints of topological structure on trajectory modeling.
- **LSTM, Bi-LSTM** [12, 23] proposes a classifier that automatically processes the features from trajectories.
- **TrajFormer, TrajODE** [19, 20] considers spatio-temporal intervals to generate trajectory embeddings.
- **SECA** [6] automatically extracts relevant features from trajectories and exploits useful information in unlabeled data.
- **DNN, TraClets** [8, 14] extracts the self-images feature and fuses it with the trajectory embedding.
- **Estimator, Estimator\*** [13] integrates the self-images with time-interval embedding and frozen partial parameters by [6].

As can be seen, an obvious trend is that Traj2Former outshines all baseline contenders. Particularly, Traj2Former shows a 10.54% enhancement over trajectory-only methods, *i.e.*, SVM, KNN, RF, RNN, GRU, LSTM, Bi-LSTM, TrajFormer, TrajODE, and SECA, for the Geolife dataset, and an 8.57% increase on the MTL dataset. This improvement suggests that the Traj2Former model adeptly captures the distribution of nearby trajectories, thereby enhancing its performance relative to models considering only pure trajectories. Specifically, in comparison to image-related methods such as DNN, TraClets, Estimator, and Estimator\*, Traj2Former exhibits advancements of at least 8.15% and 2.99% on the Geolife and MTL datasets, respectively. These figures underscore the effectiveness of the generated global map, revealing the fact that considering nearing trajectories improves the classification performance. In specific sub-classes like cars, buses, and subways within the Geolife dataset, Traj2Former achieves notable improvements. For the MTL dataset, Traj2Former significantly outperforms all baselines in all public transportation classes. Such findings demonstrate that Traj2Former has the capability to identify similar modes of transportation, by leveraging the consistent relationship between the compressed trajectory embedding and the corresponding local contexts.

## 6.3 Ablation Studies

We conduct ablation studies to evaluate the effectiveness of each design strategy. Experiments are implemented on the Geolife dataset.

*6.3.1 Impact of Different Components in Proposed Model.* Table 2 presents the effectiveness of each designed components. Firstly, the transformer-based map encoder is replaced with the CNN, resulting in the most decreased accuracy by 3.11%. This fact reveals the designed attention module can reduce the noise from global maps and focus on the local context information. Besides, we remove the trajectory compressor module but directly concatenate the trajectory embedding to the global map, leading to a 1.96% decrease. This shows the merits of the compressor module since it effectively reduces the duplicated local context features from the global map. In the experiment, *i.e.*, w/o Auxiliary Losses, we only compute the loss of the final layer and exclude the loss of the middle layers. The performance is decreased by 0.97%, demonstrating the dependency of the model's performance on its internal layer-wise interactions. This cumulative evidence underscores the effectiveness of each component in the Traj2Former model.

**Table 1: Performance comparison of Traj2Former with the state-of-the-art methods on the Geolife and MTL datasets.**

| Method | Geolife | | | | | | | |
|---|---|---|---|---|---|---|---|---|
| | Walk | Bike | Car | Bus | Subway | Train | Acc | Δ |
| KNN | 69.19% | 56.01% | 53.60% | 56.88% | 59.02% | 78.04% | 62.70% | -30.79% |
| SVM | 63.68% | 74.47% | 56.03% | 57.28% | 0.00% | 87.41% | 63.44% | -30.05% |
| RF | 73.57% | 65.57% | 57.93% | 60.08% | 64.44% | 82.00% | 67.16% | -26.33% |
| LSTM | 96.50% | 64.93% | 63.69% | 60.92% | 31.04% | 82.50% | 73.72% | -19.77% |
| GRU | 98.72% | 64.21% | 58.88% | 66.64% | 12.90% | 82.00% | 74.16% | -19.33% |
| RNN | 96.44% | 66.81% | 59.58% | 68.36% | 20.56% | 84.00% | 74.40% | -19.09% |
| Bi-LSTM | 96.05% | 60.89% | 65.26% | 71.41% | 45.56% | 84.25% | 76.59% | -16.90% |
| STGRU | 98.38% | 62.48% | 69.20% | 66.56% | 58.87% | 82.50% | 77.70% | -15.79% |
| TrajFormer | 98.22% | 81.09% | 72.79% | 71.02% | 67.33% | 92.50% | 82.80% | -10.69% |
| SECA | 96.16% | 82.25% | 75.32% | 70.63% | 74.59% | 91.00% | 82.95% | -10.54% |
| TrajODE | 99.16% | 83.11% | 72.44% | 72.82% | 68.95% | 87.75% | 83.46% | -10.03% |
| DNN | 90.05% | 67.53% | 55.29% | 62.33% | 19.75% | 78.50% | 69.77% | -23.72% |
| TraClets | 96.22% | 75.76% | 59.58% | 62.73% | 58.47% | 76.75% | 75.36% | -18.13% |
| Estimator | 98.44% | 80.51% | 75.24% | 73.06% | 68.95% | 90.75% | 83.74% | -9.75% |
| Estimator* | 96.83% | 87.01% | 74.71% | 77.52% | 74.19% | 93.00% | 85.34% | -8.15% |
| Traj2Former w/o Attention | 99.27% | 90.33% | 93.17% | 81.75% | 51.20% | 94.27% | 90.38% | -3.11% |
| Traj2Former | 99.16% | 97.11% | 87.22% | 91.15% | 94.35% | 86.50% | 93.49% | - |

| Method | MTL | | | | | |
|---|---|---|---|---|---|---|
| | Walk | Bike | Car | Public | Acc | Δ |
| KNN | 57.55% | 56.73% | 66.14% | 43.26% | 57.09% | -36.59% |
| SVM | 47.00% | 60.86% | 74.12% | 43.41% | 57.46% | -36.22% |
| RF | 67.74% | 65.63% | 69.97% | 51.12% | 64.06% | -29.62% |
| LSTM | 90.32% | 87.30% | 80.67% | 53.42% | 76.99% | -16.69% |
| GRU | 89.13% | 90.28% | 77.55% | 66.36% | 80.95% | -12.74% |
| RNN | 93.15% | 90.73% | 77.30% | 53.95% | 77.50% | -16.18% |
| Bi-LSTM | 89.58% | 85.33% | 82.89% | 35.99% | 72.51% | -21.71% |
| STGRU | 90.92% | 95.04% | 83.78% | 54.64% | 80.71% | -12.97% |
| TrajFormer | 94.34% | 95.23% | 89.44% | 63.08% | 85.11% | -8.57% |
| SECA | 95.53% | 94.98% | 83.97% | 63.47% | 83.61% | -10.07% |
| TrajODE | 95.08% | 94.79% | 84.17% | 64.76% | 83.89% | -9.79% |
| DNN | 94.69% | 90.03% | 95.86% | 21.49% | 76.12% | -17.56% |
| TraClets | 94.94% | 93.90% | 88.17% | 59.59% | 83.50% | -10.18% |
| Estimator | 95.57% | 93.06% | 81.98% | 70.76% | 84.51% | -9.17% |
| Estimator* | 95.72% | 97.98% | 86.24% | 84.44% | 90.69% | -2.99% |
| Traj2Former w/o Attention | 96.72% | 91.11% | 95.99% | 86.30% | 92.11% | -1.57% |
| Traj2Former | 93.30% | 93.58% | 94.53% | 92.99% | 93.68% | - |

*6.3.2 Comparison with Different Map Generation Strategies.* To demonstrate the effectiveness of the generated global maps, we feed different feature maps to the Traj2Former model. As shown in Table 3, using the global map consisting of physical features (*i.e.*, $C_f$ in Eq. 8) and class distribution features (*i.e.*, $C_d$ in Eq. 9) performs better than applying only the self-images by 8.10%. This fact shows the effectiveness of the generated global maps. For the components of the global maps, only using the physical maps $C_p$ performs worse by 1.73%. It demonstrates that considering the class distribution improves the performance of our model. Besides, we show the versatility of the Traj2Former model by concatenating global maps to the self-image to achieve 94.21% accuracy.

*6.3.3 Ablation Study of Compress Rate.* To evaluate the different compress rates of the Traj2Former model, we adjust the compress

rate $S_1$ of the first layer and give fixed compress rates $S_i$=2 for further layers $i$=2, $\cdots$, $L$, shown in Table 4. When we decrease the compress rate from 30 to 5, the overall trend is that the performance increases first and then decreases. In particular, reducing the compress rate first results in more local context images from the global map, and the accuracy increases. When the compress rate is increased further, the accuracy decreases. That is, a small compress rate results in duplicated local contexts, especially when we have a dense GPS distribution under a small interpolation size.

Fig. 4 demonstrates the accuracy trends for different subclasses at assorted compression rates. We observe that when the compress

**Table 2: Impact of different components in proposed model.**

| Methods | Acc | Δ |
|---|---|---|
| w/o Attention | 90.38% | -3.11% |
| w/o Compressor | 91.53% | -1.96% |
| w/o Auxiliary Losses | 92.52% | -0.97% |
| Traj2Former Full Model | 93.49% | - |

**Table 3: Comparison of different map generation strategies.**

| Self Image | Global Maps | | Acc | Δ |
|---|---|---|---|---|
| | $C_p$ | $C_d$ | | |
| √ | - | - | 85.39% | -8.10% |
| - | √ | - | 91.76% | -1.73% |
| - | √ | √ | 93.49% | - |
| √ | √ | √ | 94.21% | +0.72% |

**Table 4: Ablation study of compress rate**

| Compress Rate | Acc | Δ |
|---|---|---|
| $S_1 = 30$ | 93.49% | - |
| $S_1 = 15$ | 95.39% | +1.9% |
| $S_1 = 10$ | 95.07% | +1.58% |
| $S_1 = 6$ | 92.68% | -0.81% |
| $S_1 = 5$ | 91.27% | -2.22% |

**Figure 4: Per-class comparison with varying compress rate.**

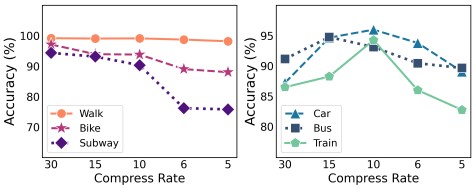

**Table 5: Impact of local context size $c_p$.**

| Local Context Size | Acc | Δ |
|---|---|---|
| $c_p$ =3 | 93.49% | - |
| $c_p$ =5 | 96.09% | +2.60% |
| $c_p$ =7 | 93.27% | -0.22% |
| $c_p$ =9 | 92.26% | -1.23% |

**Table 6: Evaluation of satellite map.**

| Map Source | | CNN | Traj2Former |
|---|---|---|---|
| Global Map | Satellite Map | | |
| √ | - | 90.69% | 93.49% |
| - | √ | 86.39% | 92.19% |
| √ | √ | 91.79% | 95.00% |

**Table 7: Impact of the filtering methods.**

| Global Map | Acc | Δ |
|---|---|---|
| CNN w/o Filter | 89.75% | -3.74% |
| CNN with Filter | 90.69% | -2.80% |
| Traj2Former w/o Filter | 92.30% | -1.19% |
| Traj2Former with Filter | 93.49% | - |

rate decreases, the accuracy of the sub-classes such as Walk, Bike, and Subway decreases. For the Car, Bus, and Train, an initial increase in accuracy is observed, followed by a subsequent decline. Accuracy diminishes for the Walk, Bike, and Subway subclasses as compression rates decrease, likely due to duplicated local contexts caused by lower speeds and Subway has a relatively constant speed. In contrast, Car, Bus, and Train initially show improved accuracy, probably of their stop-and-go moving pattern, but this benefit is lost as accuracy drops when compression rates decline excessively.

*6.3.4 Impact of Local Context Size $c_p$.* To detect the impact of the local context size $c_p$, we adjust $c_p$ from 3 to 9. The experiment results are shown in Table 5. With increasing the local context size, the accuracy increases first and then decreases. That is when expanding $c_p$, more information on the distribution of nearby trajectories is provided to distinguish different mode categories. However, this benefit is lost since further expending $c_p$ results in the distribution of all local contexts being similar and difficult to distinguish.

*6.3.5 Evaluation of Satellite Map.* To evaluate the effectiveness of our proposed Traj2Former model on the actual map dataset, we extract the actual map from the HERE map. Global maps are the original self-built maps, including physical features and class distribution features. Satellite maps are the 3-channel images from the real map. To better compare the baseline models with our proposed Traj2Former model, we implement the CNN model as the baseline model. Table 6 presents Traj2Former performing better than CNN on the global map, demonstrating the model's effectiveness. Similar results can be obtained on the satellite map dataset. When fusing the global map and satellite map, Traj2Former performs better than using the global map and satellite map individually.

*6.3.6 Impact of the Filtering Methods.* To further reduce the noise and irrelevant information in the generated global map, it is feasible to retrieve only nearby candidates around the target trajectory instead of gathering all trajectories within the self-image region. In this way, the global map is purged of noise data, thereby sharpening the relevance of the information to the target trajectory and

**Table 8: Impact of different fusion methods.**

| Methods | Acc | Δ |
|---|---|---|
| Local Fusion | 89.41% | -4.08% |
| Global Fusion | 91.53% | -1.96% |
| Multi-scale Fusion (Traj2Former) | 93.49% | - |

**Table 9: Model complexity and resource requirement.**

| Methods | Acc | Para. (M) | $\Delta T$ (s) | Local Crop (s) | Global Crop (s) |
|---|---|---|---|---|---|
| Traj2Former | 92.30% | 29.08 | 5e-2 | 0.0016 | 0.0003 |
| TraClets | 75.36% | 138.57 | 6.2e-5 | - | - |
| SECA | 82.95% | 0.33 | 3.8e-4 | - | - |
| Estimator | 83.74% | 0.57 | 7.8e-5 | - | - |

improving the quality of the global map. Table 7 presents the comparison. With the filter stage, the improvements in Traj2Former perform better than that of the CNN model, revealing that filter effectiveness can be enhanced by the Traj2Former model.

*6.3.7 Impact of Different Fusion Methods.* To demonstrate the effectiveness of our Hierarchical Global-Local Fusion module, we implement global-only and local-only fusion experiments. For the global-only fusion, we directly compress the trajectory to a segment-level embedding and fuse it with the global map embedding. The local-only fusion applies the first compress layer and fuses the outputs with the local context snapshot in our point-to-pixel fusion method. Table 8 presents that considering global or local fusion separately cannot perform better than our Global-Local Fusion method since our fusion methods address the advantages of coarse-grained feature increase and consistency in fine-grained level.

*6.3.8 Efficiency.* Table 9 reports the accuracy (Acc), model complexity (Para.), time cost per sample ($\Delta T$), and Local/Global Crop time. Noted that our method generates a complete offline map only once. Thus, the major overhead of our method is the multi-local-map cropping process, which is currently computed in a serialized manner but can be accelerated with parallel computing. The processing time per sample is 0.05 s for Traj2Former without applying filtering in the global map generation. The balance between accuracy and efficiency should be determined based on the specific application needs, *e.g.*, to enhance efficiency by increasing the compression ratio or to boost accuracy by generating a filtered global map, with a compromise on the other factor.

## 7 Conclusion

In this paper, the Traj2Former framework is designed to solve the trajectory classification problem. This framework generates global maps from the historical trajectory database to serve as image-based feature embedding and subsequently concatenated with trajectory embedding. To address the advantages of segment-to-image fusion and point-to-pixel fusion, we design a multi-scale fusion framework to generate an enhanced representation. Additionally, to improve the quality of global maps, the transformer-based map encoder is developed to focus on the alignment between the compressed trajectory and the corresponding local context snapshot. Extensive experiments are conducted on two real-world datasets to demonstrate the effectiveness of our developed framework.

## Acknowledgments

This research is partially supported by Singapore Ministry of Education Academic Research Fund Tier 2 under MOE's official grant number T2EP20221-0023. This work is partially supported by the Program of NSFC (Grant No. 62172157) and the Programs of Hunan Province (Grant Nos. 2024JJ2026, 2023GK2002).

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
