# OpenReview forum: "Traj2Former: A Local Context-aware Snapshot and Sequential Dual Fusion Transformer for Trajectory Classification"
_acmmm.org/ACMMM/2024/Conference — MM2024 Poster_

### Official Review · Reviewer_asJn · 2024-05-19

**Rating:** 3
**Confidence:** 3

**Summary:**

The authors propose a novel model termed Traj2Former to address challenges in trajectory classification. The model aims to improve the detection of nuanced variances in trajectory classification tasks by leveraging local contextual information and enhancing the integration between trajectory data and spatial contexts. This paper incorporates a new GPS rendering method to generate contextual snapshots and employs a multi-scale sequential fusion to capture diverse temporal patterns. The model is designed to be adaptable to various context sources, such as trajectory databases, digital maps, or satellite imagery. Extensive experiments demonstrate that Traj2Former achieves state-of-the-art classification accuracy on two real-world datasets, Geolife and MTL.

**Strengths:**

1: The paper proposes a unified fusion framework, which effectively combines trajectory and map data, leveraging both point-level and segment-level embeddings to capture complex spatial and temporal patterns.

2: The model's ability to work with various context sources (trajectory databases and digital maps) enhances its applicability across different domains and scenarios.

3: The paper conducts sufficient ablation studies to evaluate the effectiveness of each design strategy.

**Limitations:**

1: Although the model shows superior performance, it lacks comparison with more extensive and state-of-the-art baselines comparison as follows.

[1]. TIF（2023）—— “Cristiano Landi, Riccardo Guidotti, Mirco Nanni, and Anna Monreale. 2023. The Trajectory Interval Forest Classifier for Trajectory Classification. In Proceedings of the 31st ACM International Conference on Advances in Geographic Information Systems (SIGSPATIAL ’23). Article 67, 4 pages.”

[2]. TrajODE（2021）—— “Liang Y, Ouyang K, Yan H, et al. Modeling Trajectories with Neural Ordinary Differential Equations[C]//IJCAI. 2021: 1498-1504.”

[3]. CIF（2020）—— “Middlehurst M, Large J, Bagnall A. The canonical interval forest (CIF) classifier for time series classification[C]//2020 IEEE international conference on big data (big data). IEEE, 2020: 188-195.”

[4]. ST-GRU（2019）—— “Liu H, Wu H, Sun W, et al. Spatio-temporal GRU for trajectory classification[C]//2019 IEEE International Conference on Data Mining (ICDM). IEEE, 2019: 1228-1233.”

2: While the model performs well on the Geolife and MTL datasets, it is suggested to conduct more experiments on another two datasets: TDrive and NYCTaxi, to see the generality of the proposed method.

3: The proposed model's complexity might pose challenges for practical implementation, the paper lacks experiments related to computational resources and efficiency required for model training.

**Suitability:**

2

---

### Official Review · Reviewer_KTpi · 2024-05-20

**Rating:** 3
**Confidence:** 3

**Summary:**

The paper proposes Traj2Former,  which addresses challenges in trajectory classification arising from noisy and sparse GPS data. The methodology is robust, with a clear explanation of the dual fusion mechanism. The experimental results are impressive, showing clear benefits over traditional models. However, it needs some revisions based on the limitation part.

**Strengths:**

1. The paper is well structured, clearly explaining the motivation, challenges, proposed architecture and experimental findings.
2. The design decisions of the architecture are well explained and supported with strong ablation studies.
3.The strong baselines including recent methods are provided.
4. The code is released which helps the community to easily build upon the contributions.

**Limitations:**

1. The author did not specify the settings for the sliding window in the text, which I consider a critical hyperparameter. Specific experiments or descriptions should be added.
2. Section 4.1.2 does not clearly define what function or method f_emb refers to, the author should provide a more detailed explanation.
3. The class ratio in the appendix is crucial, it provides further insights into the experimental results and should be moved to the main text.
4. Table 1 shows that Traj2Former without attention performs poorly on the subway category in the Geolife dataset, with only a 51.2% accuracy. Why is this the case?
5. In Table 1, should the 'car' category in the MTL dataset be labeled as 'bus' based on the information in the appendix? Is this a typographical error?
6. An odd phenomenon is that, in the MTL dataset, there are more data points for the Bike category than for the Walk category, yet the Traj2Former performs worse on Bike and is weaker than the state of the art. Normally, with more data, the model should learn more and thus perform better.
7. Regarding the ablation experiments, some settings in Section 6.3.2-5 perform better than the best metrics in Table 1, which are 93.49%. Specifically, why isn't the best ACC result in Table 1 the highest at 96.09% on Table 5?

**Suitability:**

3

---

### Official Review · Reviewer_FueG · 2024-05-24

**Rating:** 4
**Confidence:** 2

**Summary:**

The paper introduces a novel model, Traj2Former,  for trajectory classification using deep learning techniques. Generating contextual snapshots to capture the spatial distribution of nearby trajectories. Implementing a multi-scale sequential fusion technique to compress trajectory data and capture diverse temporal patterns. Extensive experiments demonstrate that Traj2Former achieves state-of-the-art classification accuracy on two real-world datasets, validating its effectiveness in trajectory classification tasks.

**Strengths:**

1. The paper introduces a novel framework, Traj2Former, which combines local context-aware snapshots with a multi-scale sequential dual fusion transformer for trajectory classification. The approach of generating contextual snapshots and using a transformer-based map encoder to handle noise and refine local contexts is innovative.  Previous methods either did not fully explore the spatial distribution of nearby trajectories or failed to model the local correspondence between GPS points and image pixels. Traj2Former bridges this gap by integrating both local and global spatial contexts, providing a more holistic view of the trajectory data.

2. The use of a multi-scale sequential fusion technique to compress trajectory data at different rates is theoretically robust. This method allows for capturing diverse temporal patterns, which is critical for accurate trajectory classification. The transformer-based map encoder is a technically sound choice for handling large-scale data with noise and sparsity. The attention mechanisms used in transformers are well-suited for identifying relevant spatial features in the context of trajectory data.

3. The paper is well-organized and clearly written, making it easy to follow the methodology and understand the contributions.
Figures and tables are used effectively to illustrate the model architecture, the experimental setup, and the results.

**Limitations:**

1. Some components of the model, such as the use of transformer-based encoders and multi-scale fusion, are well-established techniques in the field of deep learning and computer vision. These methods have been widely used in various other contexts, which may reduce the perceived novelty of the paper. Although the specific application of these techniques to trajectory classification is innovative, the individual components themselves are not new. For instance, transformers and attention mechanisms are standard in many current deep learning models (Vaswani et al., 2017).

2. The paper focuses heavily on comparisons with other deep learning models but provides limited evaluation against traditional machine learning methods. This could give a skewed perspective of the model's performance. Including comparisons with well-known traditional methods, such as SVMs, decision trees, or KNN, would provide a more comprehensive understanding of the improvements offered by Traj2Former.

3. The evaluation is conducted on only two datasets, Geolife and MTL. While these are valuable datasets, the model’s robustness and generalizability could be better demonstrated with a more diverse set of datasets from different domains or geographic regions.

4. The proposed model involves multiple complex components, such as the transformer-based map encoder and the multi-scale sequential fusion, which might be computationally intensive and challenging to scale for very large datasets. Although not directly addressed in the paper, the computational requirements for training and deploying Traj2Former could be significant. A discussion on the model's scalability, resource requirements, and potential optimization strategies would be beneficial.

5. While the paper provides a clear high-level overview, some of the implementation details are sparse. For example, specific hyperparameters, the exact structure of the transformer layers, and the computational costs are not thoroughly discussed. Detailed implementation information is crucial for replicability and practical application of the research. Including more comprehensive details would help practitioners and researchers to better understand and implement the model.

**Suitability:**

2

---

### Official Review · Reviewer_vvAG · 2024-05-26

**Rating:** 3
**Confidence:** 3

**Summary:**

The paper introduces Traj2Former, which leverages both local context and sequential information for trajectory classification. The authors have identified key issues in existing methods, such as the inability to effectively utilize the distribution of nearby trajectories and the loss of local correspondence information when trajectories are rendered into images.

**Strengths:**

1. The paper presents a transformer-based model that considers both the local context and the sequential nature of trajectory data.

2. The idea of generating contextual snapshots from a global map to provide critical contextual features is new enhances the model's ability to classify trajectories accurately.

3. This paper conducted comprehensive experiments on two real-world datasets, demonstrating the superiority of Traj2Former over existing state-of-the-art methods.

**Limitations:**

1. The paper is lack of discussion of the computational complexity of the proposed method, especially considering the use of transformer models which can be computationally intensive.

2. While the method shows promising results on the tested datasets, it would be beneficial to evaluate its performance on other types of trajectory data and across different domains.

**Suitability:**

2

---

### Meta-Review · Area_Chair_htDP · 2024-07-04

**Recommendation:** Accept (Poster)
**Confidence:** 4

**Metareview:**

The authors propose Traj2Former, a framework for trajectory classification. In particular, this framework has the goal of improving trajectory classification tasks by using local contextual information and integrating trajectory data and spatial contexts. As noticed by one of the reviewer, the paper incorporates an interesting GPS rendering method to generate contextual snapshots as well as a multi-scale sequential fusion methodology to capture diverse temporal patterns. The experimental work is solid and conducted extensively on two datasets, Geolife and MTL, in the original paper and on more datasets in the rebuttal. Also the comparisons with other approaches are adequate: various baseline and alternative models were tested in the original paper and in the rebuttal. However, as pointed out by Reviewer vvAG, several of the components of the models are well established one in computer vision and also multimedia communities, so the suggestion is to highlight better the novel aspects of the proposed approach. Finally, the authors conducted a detailed rebuttal where they target the various concerns raised by the reviewers.